# Discriminative Cross-Modal Data Augmentation for Medical Imaging Applications

## Abstract

While deep learning methods have shown great success in medical image analysis, they require a number of medical images to train. Due to data privacy concerns and unavailability of medical annotators, it is oftentimes very difficult to obtain a lot of labeled medical images for model training. In this paper, we study cross-modality data augmentation to mitigate the data deficiency issue in the medical imaging domain. We propose a discriminative unpaired image-to-image translation model which translates images in source modality into images in target modality where the translation task is conducted jointly with the downstream prediction task and the translation is guided by the prediction. Experiments on two applications demonstrate the effectiveness of our method.

## 1 Introduction

Developing deep learning methods to analyze medical images for decision-making has aroused much research interest in the past few years. Promising results have been achieved in using medical images for skin cancer diagnosis (Esteva et al., 2017; Tschandl et al., 2019), chest diseases identification (Jaiswal et al., 2019), diabetic eye disease detection (Cheung et al., 2019), to name a few. It is well-known that deep learning methods are data-hungry. Deep learning models typically contain tens of millions of weight parameters. To effectively train such large-sized models, a large number of labeled training images are needed. However, in the medical domain, it is very difficult to collect labeled training images due to many reasons including privacy barriers, unavailability of doctors for annotating disease labels, etc.

To address the deficiency of medical images, many approaches (Krizhevsky et al., 2012; Cubuk et al., 2018; Takahashi et al., 2019; Zhong et al., 2017; Perez & Wang, 2017) have been proposed for data augmentation. These approaches create synthetic images based on the original images and use the synthetic images as additional training data. The most commonly used data augmentation approaches include crop, flip, rotation, translation, scaling, etc. Augmented images created by these methods are oftentimes very similar to the original images. For example, a cropped image is part of the original image. In clinical practice, due to the large disparity among patients, the medical image of a new patient (during test time) is oftentimes very different from the images of patients used for model training. If the augmented images are very close to the original images, they are not very useful in improving the ability of the model to generalize to unseen patients. It is important to create diverse augmented images that are non-redundant with the original images.

To create non-redundant augmented images for one modality such as CT, one possible solution is to leverage images from other modalities such as X-ray, MRI, PET, etc. In clinical practice, for the same disease, many different types of imaging techniques are applied to diagnose and treat this disease. For example, to diagnose lung cancer, doctors can use chest X-rays, CT scans, MRI scans, to name a few. As a result, different modalities of medical images are accumulated for the same disease. When training a deep learning model on an interested modality (denoted by $X$) of images, if the number of original images in this modality is small, we may convert the images in other modalities into the target modality $X$ and use these converted images as additional training data. For example, when a hospital would like to train a deep learning model for CT-based lung cancer diagnosis, the hospital can collect MRI, X-ray, PET images about lung cancer and use them to augment the CT training dataset. Images of different modalities are typically from different patients. Therefore, their clinical diversity is large.

This motivates us to study cross-modality data augmentation to address the data deficiency issue in training deep learning models for medical image based clinical decision-making. The problem setup is as follows. The goal is to train a deep learning model to diagnose disease $D$ based on one modality (denoted by $X$) of medical images. However, the number of images in this modality is limited, which are not sufficient to train effective models. Meanwhile, there is another modality (denoted by $Y$) of medical images used for diagnosing disease $D$. Cross-modality data augmentation refers to leveraging the images in $Y$ to augment the training set in $X$. Specifically, we translate images in $Y$ into images that have a similar style as those in $X$ and add the translated images together with their disease labels into the training dataset in $X$. Compared with simple augmentation such as cropping, scaling, rotation, cross-modality data augmentation can bring in more diversity since the images in different modalities are from different patients and hence are clinically more heterogeneous. More diverse augmented images are more valuable in improving the generalization ability of the model to unseen patients.

To perform cross-modality data augmentation, we propose a discriminative unpaired image-to-image translation (DUIIT) method. Given images in a source modality, we translate them into images in a target modality. The translated images, together with their associated disease labels, are added to the training set in the target modality to train the predictive model. Different from unsupervised translation methods such as CycleGAN (Zhu et al., 2017a) which perform the translation between images without considering their disease labels, our method conducts the translation in a discriminative way, where the translation is guided by the predictive task. Our model performs cross-modality image translation and predictive modeling simultaneously, to enable these two tasks to mutually benefit each other. The translated images are not only aimed to have similar style as those in the target modality, but also are targeted to be useful in training the predictive model.

Our model consists of two modules: a translator and a predictor. The translator transforms the images in the source modality into synthesized images in the target modality. Then the translated images (together with their class labels) are combined with the images in the target modality to train the predictor. The predictor takes an image as input and predicts the class label. The two modules are trained jointly so that the translator is learned to generate target images that are effective in training the predictor.

We apply our method to two medical imaging applications. In the first application, the source modality is MRI and the target modality is CT. Our method translates MRI images into CTs in a discriminative way and uses the combination of original CTs and translated CTs to train the predictive model, which achieves substantially lower prediction error compared with using original CTs only. In the second application, the source modality is PET and the target modality is CT. By translating PET images to CTs, our method significantly improves prediction performance.

The major contributions of this paper include:

- We propose a discriminative unpaired image-to-image translation method to translate medical images from the source modality to the target modality to augment the training data in the target modality. The translation is guided by the predictive task.

- On two applications, we demonstrate the effectiveness of our method.

The rest of the paper is organized as follows. Section 2 reviews related works. Section 3 and 4 present the methods and experiments. Section 5 concludes the paper.

## 2  RELATED WORKS

### 2.1  MEDICAL IMAGE SYNTHESIS

Several attempts have been made to synthesize medical images. Frid-Adar et al. (2018) combine three DCGANs (Radford et al., 2015) to generate synthetic medical images, which are used for data augmentation and lesion classification. Jin et al. (2018) develop a 3D GAN to learn lung nodule properties in the 3D space. The 3D GAN is conditioned on a volume of interest whose central part containing nodules has been erased. GANs are also used for generating segmentation maps (Guibas et al., 2017) where a two stage pipeline is applied. Mok & Chung (2018) apply conditional GANs (Mirza & Osindero, 2014; Odena et al., 2017) to synthesize brain MRI images in a coarse-to-fine manner, for brain tumour segmentation. To reserve fine-grained details of the tumor core, they encourage the generator to delineate tumor boundaries. Cross-modality translation has

been studied to synthesize medical images. In Wolterink et al. (2017), it is found that using unpaired medical images for augmentation is better than using aligned medical images. Chartsias et al. (2017) apply CycleGAN for generating synthetic multi-modal cardiac data. Zhang et al. (2018) combine the synthetic data translated from other modalities with real data for segmenting multimodal medical volumes. A shape-consistency loss is used to reduce geometric distortion.

## 2.2 IMAGE GENERATION

Generative Adversarial Networks (GANs) (Goodfellow et al., 2014; Radford et al., 2015; Arjovsky et al., 2017) have been widely used for image generation from random vectors. Conditional GANs (Mirza & Osindero, 2014; Odena et al., 2017) generates images from class labels. Image-to-image translation (Isola et al., 2016; Wang et al., 2018; Zhu et al., 2017a) studies how to generate one image (set) from another (set) based on GANs. A number of works have been devoted to generating images from texts. Mansimov et al. (2015) propose an encoder-decoder architecture for text-to-image generation. The encoder of text and the decoder of image are both based on recurrent networks. Attention is used between image patches and words. AttnGAN (Xu et al., 2018) synthesizes fine-grained details at different subregions of the image by paying attention to the relevant words in the natural language description. DM-GAN (Zhu et al., 2019) uses a dynamic memory module to refine fuzzy image contents, when the initial images are not well generated, and designs a memory writing gate to select the important text information. Obj-GAN (Li et al., 2019) proposes an object-driven attentive image generator to synthesize salient objects by paying attention to the most relevant words in the text description and the pre-generated semantic layout. MirrorGAN (Qiao et al., 2019) uses an autoencoder architecture, which generates an image from a text, then reconstructs the text from the image. Many techniques have been proposed to improve the fidelity of generated images by GANs. Brock et al. (2018) demonstrate that GANs benefit remarkably from scaling: increasing model size and minibatch size improves the fidelity of generated images. To generate images with higher resolution, the progressive technique used in PGGAN (Karras et al., 2017) has been widely adopted. StackGAN (Zhang et al., 2017) first uses a GAN to generate low-resolution images, which are then fed into another GAN to generate high-resolution images.

## 2.3 IMAGE AUGMENTATION

Image augmentation is a widely used technique to enlarge the training dataset and alleviate overfitting. Basic augmentation methods (Krizhevsky et al., 2012) include geometric and color transformations, such as crop, flip, rotation, translation, scale, color jitter, contrast, etc. Cubuk et al. (2018) propose a reinforcement learning based algorithm to automatically search for augmentation policies. Takahashi et al. (2019) propose to randomly crop four images and patch them to create a new training image. Zhong et al. (2017) propose Random Erasing which assigns random values to pixels in randomly-sampled rectangle regions, to generate augmented images with different occlusion levels. Perez & Wang (2017) design two methods for data augmentation. One is using GAN to synthesize new images with different styles. The other is training an augmentation network for generating augmented data. Zhao et al. (2020) propose DiffAugment that can improve the sample efficiency of training GANs. This method applies the same differentiable augmentation to both real and fake images when training the generator and discriminator.

## 3 METHOD

In this section, we introduce a discriminative unpaired image-to-image translation method for cross-modal data augmentation. We are given a set of images in modality $X$ and another set of images in modality $Y$. For example, $X$ could be CTs and $Y$ could be MRIs. For each image in $X$ and $Y$, it is associated with a class label $r \in \mathcal{R}$, where the label space $\mathcal{R}$ is shared by images in these two modalities. Our goal is to learn a predictive model which maps images in $X$ to $\mathcal{R}$. However, the number of training images in $X$ is limited. Therefore, we would like to use images in $Y$ to augment the training set in $X$. The basic idea is: we develop a model which translates each image $y$ in $Y$ into an image $\hat{y}$ where the modality of $\hat{y}$ is aimed to be $X$. Then we add the translated images together with their associated class labels into the training set in $X$ and use the combined training set to train the predictive model. One way to perform such a translation is to use unpaired image-to-image translation methods such as CycleGAN (Zhu et al., 2017b). However, CycleGAN performs translation in an unsupervised way. The translated images may not be optimal for the predictive task. To address this issue, we propose a discriminative unpaired image-to-image translation approach. In our model, two tasks are performed simultaneously: unpaired image-to-image translation and

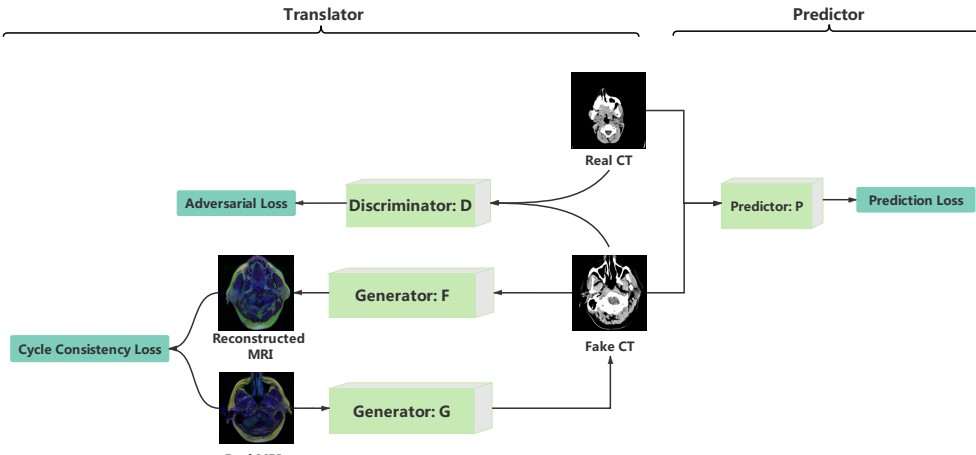

Figure 1: The architecture of our model. The source modality and target modality is CT and MRI respectively. For clarity, we omit the adversarial loss in translating CTs to MRIs.

predictive modeling. The translated images are not only encouraged to be similar to the images in $X$, but also useful to train the predictive model. Figure 1 illustrates this idea.

Our model consists of two modules: a translator and a predictor. The translator takes images in the source modality as inputs and translates them into images that are similar to those in the target modality. The predictor takes the original target images and translated images as inputs and predicts their class labels. The translator and the predictor are trained jointly end-to-end. The translated images are required to 1) be visually similar to those in the target modality; 2) informative for training the predictor. In the sequel, we introduce these two modules in detail.

### 3.1 TRANSLATOR

The translator is based on CycleGAN. Generative adversarial network (GAN) Goodfellow et al. (2014) is a deep generative model. It consists of a generator and a discriminator. The generator takes a random vector as input and generates an image. The discriminator takes an image (either real or generated) as input and predicts whether this image is real or generated. The generator and discriminator are learned jointly by solving a min-max problem where the loss is a classification loss in the task of classifying an image as real or generated. The discriminator aims to minimize this loss and the generator aims to maximize this loss. Intuitively, the discriminator tries to tell apart generated images from real ones while the generator tries to make the generated images as realistic as possible in a way that they are not distinguishable from real images.

GAN generates images from random vectors. These random vectors do not have class labels. As a result, the generated images do not have class labels. Therefore, they are not very useful for training supervised models. In our task, images in the source modality are associated with class labels. Given a source image $y$ and its class label $r$, we can translate $y$ into the target modality $\hat{y}$. Then we obtain a labeled pair $(\hat{y}, r)$ in the target domain and use this pair to train the target predictor.

However, the images in source modality and those in the target modality do not have one-to-one correspondence. Namely, a source image and a target image are from different patients. Therefore, we cannot perform the translation by training a model which maps a source image to a target image. To address this problem, we use CycleGAN (Zhu et al., 2017a), to perform unpaired image-to-image translation which translates one set of images to another set. Let $\mathcal{G}$ denote a conditional generative model which takes a source image as input and generates an image in the target modality. Let $\mathcal{F}$ denote a conditional generative model which takes a target image as input and generates an image in the source modality. Let $\mathcal{D}_s$ denote a discriminator which judges whether an image in the source modality is real or fake. Let $\mathcal{D}_t$ denote a discriminator which judges whether an image in the target modality is real or fake. CycleGAN employs a cycle-consistency loss: given a source image $y$, it is first translated into the target modality using $\mathcal{G}$: $\mathcal{G}(y)$, then $\mathcal{G}(y)$ is translated back to the source modality using $\mathcal{F}$: $\mathcal{F}(\mathcal{G}(y))$, and $\mathcal{F}(\mathcal{G}(y))$ is encouraged to be close to the original image $y$. Similarly, the cycle consistency loss can be defined on a target image $x$ as well, where $\mathcal{G}(\mathcal{F}(x))$ is

encouraged to be close to $x$. The overall loss is defined as:

$$\mathcal{L}_{trans}(\mathcal{G}, \mathcal{F}, \mathcal{D}_t, \mathcal{D}_s) = \mathcal{L}_{adv}(\mathcal{D}_t, \mathcal{G}) + \mathcal{L}_{adv}(\mathcal{D}_s, \mathcal{F})$$
$$+ \lambda_{cyc}(\mathbb{E}_{P_{data}(y)}[\|\mathcal{F}(\mathcal{G}(y)) - y\|_1] + \mathbb{E}_{P_{data}(x)}[\|\mathcal{G}(\mathcal{F}(x)) - x\|_1]) \quad (1)$$

where $\mathcal{L}_{adv}$ denotes the adversarial training loss in GAN and $\lambda_{cyc}$ is a tradeoff parameter.

### 3.2 PREDICTOR

Given $N_s$ labeled source images $\{(y_i, r_i^{(s)})\}_{i=1}^{N_s}$ where $y_i$ is an image and $r_i^{(s)}$ is its label, we translate $y_i$ into the target modality: $\mathcal{G}(y_i)$, and obtain a collection of translated images with labels: $\{(\mathcal{G}(y_i), r_i^{(s)})\}_{i=1}^{N_s}$. Then we combine these generated pairs with the training data $\{(x_i, r_i^{(t)})\}_{i=1}^{N_t}$ in the target modality and use the combined data to train a predictor. Let $\mathcal{P}$ denote the predictor and $l(\mathcal{P}(x), r)$ denote the loss function defined on a training pair $(x, r)$. Then the discriminative loss is:

$$\mathcal{L}_{pred}(\mathcal{G}, \mathcal{P}) = \sum_{i=1}^{N_s} l(\mathcal{P}(\mathcal{G}(y_i)), r_i^{(s)}) + \sum_{i=1}^{N_t} l(\mathcal{P}(x_i), r_i^{(t)}) \quad (2)$$

### 3.3 OBJECT FUNCTIONS

In our method, the translation task and the prediction task are performed jointly by optimizing the following objective function:

$$\mathcal{L}(\mathcal{G}, \mathcal{F}, \mathcal{D}_s, \mathcal{D}_t, \mathcal{P}) = \mathcal{L}_{trans}(\mathcal{G}, \mathcal{F}, \mathcal{D}_s, \mathcal{D}_t) + \lambda \mathcal{L}_{pred}(\mathcal{G}, \mathcal{P}) \quad (3)$$

where $\lambda$ is a tradeoff parameter.

## 4 EXPERIMENT

In this section, we present experimental results. The target modality is CT and the clinical task is to predict the physiological age of a patient based on his or her CT image. Physiological age (Wang et al., 2017) is a measure of how well or poorly one's body is functioning. It may be different from the chronological age (which is calculated based on birth

Table 1: Statistic of the three datasets.

|     | Image number | Image size | Patient number |
|-----|--------------|------------|----------------|
| MRI | 3454         | $256 \times 256$ | 110 |
| CT  | 2438         | $650 \times 650$ | 82  |
| PET | 12780        | $128 \times 128$ | 298 |

date). For example, a 30-year-old young person may have a physiological age of 40 if his or her biological system is not functioning well. Predicting physiological age has important clinical applications in disease prognosis and treatment. We use images from two source modalities including MRI and PET to augment the CT training set. We translate MRI and PET images into CTs to help with the physiological age prediction task on CT. Each image in the three modalities is labeled with a physiological age. We use mean squared error to measure predictive performance.

### 4.1 DATASETS

Three datasets are used in our experiments: Brain MRI (Buda, 2019), Brain CT (Hssayeni, 2019; Hssayeni et al., 2020), and Head PET (Vallières et al., 2017; Vallieres et al., 2017; Clark et al., 2013). Brain MRI contains 3454 brain MRI images of size $256 \times 256$ from 110 patients. MRI stands for magnetic resonance imaging, which is a medical imaging technique used in radiology to form pictures of the anatomy and the physiological processes of the body. Brain CT contains 2438 brain CT slices of size $650 \times 650$ from 82 patients. CT stands for computed tomography, which is another type of medical imaging technique. On average, each patient has about 30 slices. Head PET (Vallières et al., 2017; Vallieres et al., 2017; Clark et al., 2013) contains 12780 head PET images of size $128 \times 128$ from 298 patients. PET stands for positron emission tomography, which is a type of nuclear medicine procedure that measures metabolic activity of the cells of body tissues. Each image in the three dataset is labeled with the physiological age of the patient. Table 1 shows the statistics of the three datasets. From the original CT dataset, we randomly select 1887 images for training, 271 images for validation, and 280 images for testing. Synthetic CTs translated from MRIs and PETs are used for training. We perform the random splits 10 times and report the mean and standard deviation of performance numbers obtained from the 10 runs.

## 4.2 IMPLEMENTATION DETAILS

In the translator, the architectures of the generator and discriminator are the same as those in Cycle-GAN (Zhu et al., 2017b). The generator consists of several convolution layers with stride 2, residual blocks (He et al., 2016) and fractionally strided convolutions with stride $\frac{1}{2}$. Different blocks are used for images with different resolution. As for the discriminator, we use the architecture of the $70 \times 70$ PatchGANs (Isola et al., 2017; Ledig et al., 2017; Li & Wand, 2016). For the predictor, we use ResNet50 (He et al., 2016) as the backbone. For the translator, we follow the same training setting as CycleGAN (Zhu et al., 2017b). We use the least-square loss (Mao et al., 2017) for $\mathcal{L}_{adv}$. Discriminators are updated using images from image buffer (Shrivastava et al., 2017). $\lambda_{cyc}$ in Equation 1 is set to 10. We use Adam (Kingma & Ba, 2014) as the optimizer. Networks in the translator are trained from scratch with a learning rate of 0.0002, which linearly decays from epoch 100. The learning rate is set to 0.001 for the predictor and we adopt the same decay strategy as for the translator. $\lambda$ in Equation 3 is set to 0.001.

## 4.3 BASELINES

We compare with the following baselines.

- **Transfer learning (TL)**. We first train the prediction network using images and their labels in the source modality. Then we transfer the learned network to the target modality. We use the network trained on source images to initialize the network for the target task, then finetune this network using images and labels in the target modality.

- **Multi-task learning (MTL)**. We develop a single network to perform the prediction task in the source modality and target modality simultaneously. The two tasks share the same visual feature learning network. A source-specific predictive head is used to make predictions on the source images and a target-specific predictive head is used to make predictions on the target images. After training, the source head is discarded. The representation learning network and the target head are retained to form the final network.

- **Domain adaptation (DA)**. We treat the source and target modality as source and target domain respectively and perform adversarial domain adaptation using the Domain-Adversarial Neural Network (DANN) (Ganin & Lempitsky, 2015) method. DANN aims to learn a feature extractor in a way that the representations of images in the source domain are indistinguishable from the target domain, for the purpose of minimizing the discrepancy of these two domains.

- **CycleGAN**. We use CycleGAN (Zhu et al., 2017b) to perform unsupervised translation of source images into the target modality. In the translation process, the age labels are not leveraged. After translation, the translated images are combined with real images for training the predictive model.

- **Data augmentation methods.** We compare with three data augmentation methods including Simple Augment, AutoAugment (Cubuk et al., 2018), and Random Erasing (Zhong et al., 2020). These methods are applied to the training CTs to create augmented CTs which are used as additional training data. In Simple Augment, we apply commonly used traditional data augmentation methods, such as crop, flip, translation, rotating, and color jitter. The augmentation operations in AutoAugment are automatically searched using reinforcement learning. Random Erasing randomly samples a rectangle from an image and replaces pixels in this rectangle with random values.

## 4.4 RESULTS

Table 2 shows the prediction errors under four settings:

- **PURE-CT**. The predictive model is trained purely on original CT images.

- **MIX-CT-MRI**. The predictive model is trained on CT images and MRI images. The MRI images are translated into synthesized CT images, which are combined with the original CT images to train the predictive model. The translation from MRI to CT is performed jointly with the training of the predictive model.

- **MIX-CT-PET**. The setting is similar to MIX-CT-MRI, except the source modality is PET.

- **MIX-CT-MRI-PET**. The setting is similar to MIX-CT-MRI, except that CT and PET are both used as source modalities and are translated into CTs.

As can be seen from this table, the mean prediction errors under MIX-CT-MRI, MIX-CT-PET, and MIX-CT-MRI-PET are much lower than that under PURE-CT. This demonstrates that it

is effective to translate images from other modalities (i.e., MRI, PET) into the target modality (i.e., CT) as augmented data for model training. Our approach is effective in performing such a translation. MIX-CT-MRI-PET performs better than MIX-CT-MRI and MIX-CT-PET because MIX-CT-MRI-PET translates more source images for training the predictive model.

Another observation is: the standard deviation of prediction errors under MIX-CT-MRI, MIX-CT-PET, and MIX-CT-MRI-PET are much lower than that under PURE-CT. This is because with cross-modal translation, MIX-CT-MRI, MIX-CT-PET, and MIX-CT-MRI-PET have more training data than PURE-CT. Increasing training data can reduce the variance of the model.

Table 2: Mean and standard deviation (std) of prediction errors under different data settings.

|  | PURE-CT | MIX-CT-MRI | MIX-CT-PET | MIX-CT-MRI-PET |
|---|---|---|---|---|
| Mean | 103.88 | 76.91 | 75.75 | 71.37 |
| Std | 64.27 | 8.31 | 20.52 | 18.92 |

Table 3: Comparison between our method and baselines

| | MRI → CT | | | | | PET → CT | | | | |
|---|---|---|---|---|---|---|---|---|---|---|
| Method | TL | MTL | DA | CycleGAN | Our method | TL | MTL | DA | CycleGAN | Our method |
| Mean | 91.52 | 99.26 | 88.13 | 147.06 | 76.91 | 102.08 | 92.94 | 148.72 | 96.70 | 75.75 |
| Std | 13.88 | 22.12 | 21.86 | 79.89 | 8.31 | 15.37 | 28.26 | 40.16 | 42.44 | 20.52 |

Table 4: Comparison between our method and data augmentation methods

|  | Simple Augment | AutoAugment | Random Erasing | Our method (MRI→CT) | Our method (PET→CT) |
|---|---|---|---|---|---|
| Mean | 96.80 | 84.65 | 84.87 | 76.91 | 75.75 |
| Std | 26.00 | 15.28 | 8.55 | 8.31 | 20.52 |

Table 3 compares our method with baselines including transfer learning (TL), multi-task learning (MTL), domain adaptation (DA), and CycleGAN. As can be seen, our method outperforms the four baselines. The reason that our method outperforms TL, MTL, and DA is because these three baseline methods do not explicitly translate source images into the target modality whereas our method performs this explicit translation. In TL, the source images are used to pretrain the prediction network. In MTL, the source images are used to train the prediction network jointly with target images. In these two methods, the source images are directly used to train networks without translation. Since the source images have large modality discrepancy with target images, the networks trained by source images may not be suitable for making predictions on target images. Domain adaptation (DA) partially addresses this problem by making the visual representations of source images and target images to be close. The adaptation is performed in the latent space which may not capture the fine-grained details at the pixel level which are crucial for making accurate clinical predictions. In contrast, our method produces raw synthetic images where the clinical details are preserved. The reason that our method outperforms CycleGAN is because the source-to-target translation in our

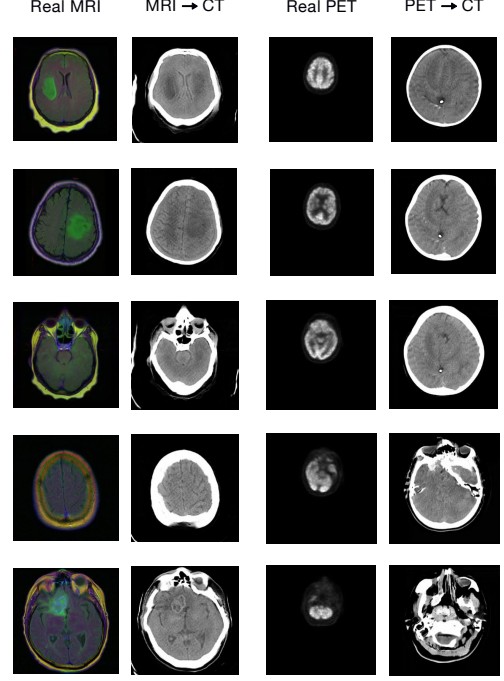

Figure 2: Examples of translating MRI and PET images into CTs.

method is discriminative and supervised by age labels, whereas CycleGAN performs the translation in an unsupervised way without leveraging the age labels. While the translated images by CycleGAN visually look like CTs, they may not be optimal for predicting the chronological ages.

Table 4 compares our method with several data augmentation methods including Simple Augment, AutoAugment, and Random Erasing. As can be seen, our method under two settings – translating MRIs to CTs and translating PETs to CTs – both performs better than the three augmentation methods. The three augmentation methods create augmented images from the training CTs. As a result, the augmented CTs are similar to training CTs with high redundancy. These augmented CTs do not bring in significantly new signals into the training set and hence are not substantially helpful in improving the generalization performance. In contrast, our method translates real MRIs and PETs into CTs. These MRIs and PETs are from other patients and contain clinical traits that are significantly different from those in the training CTs. The CTs translated from MRIs and PETs bring in substantial diversity to the training set and the model trained using them has better generalization ability.

Table 5: Evaluation of image quality using IS and FID

|  | IS ↑ | FID ↓ |
|---|---|---|
| CycleGAN | $3.49 \pm 0.15$ | **40.94** |
| Ours | $\mathbf{3.92 \pm 0.17}$ | 63.17 |

We evaluate the quality of translated images. Figure 2 shows some examples of translating MRI and PET images into CTs by our method. As can be seen, these translated images look like real CTs. Many fine-grained details in the original MRI/PET images are well-preserved in the translated images. These details are important for correctly predicting physiological ages. Table 5 compares the inception score (IS) (Salimans et al., 2016) and Frechet inception distance (FID) (Heusel et al., 2017) achieved by CycleGAN and our method. Figure 3 shows the comparison between images generated by Cycle-GAN and our method. As can be seen, our method achieves a higher (better) inception score. Inception score measures the realisticness and diversity of generated images. This result demonstrates that the images generated by our method are more realistic and diverse. The reason is that our method translates images in a discriminative way, by encouraging the translated images to be suitable for the prediction task. In contrast, CycleGAN performs the translation purely based on style matching and ignores the supervised information. On the other hand, the FID achieved by our method is larger (worse) compared

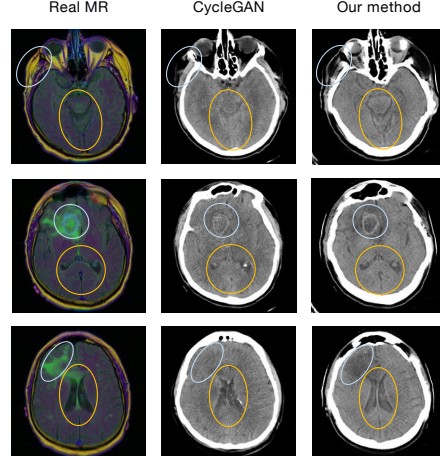

Figure 3: Some examples of translating MRIs to CTs by CycleGAN and our method. As shown in the regions marked with ovals, our method can better preserve fine-grained details in the translated images than Cycle-GAN.

to CycleGAN. FID measures the similarity between generated images and target images. This result demonstrates that images generated by our method are less similar to the target images compared with CycleGAN. The reason is that our method retains details of some soft tissues that are contained in source images but not in target images. These details render the translated CTs are more different from the real CTs. However, this is not necessarily a disadvantage. In our method, the goals of translation are two-fold: 1) the translated images are visually similar to those in the target modality; 2) the translated images preserve important clinical details that are informative for predicting chronological ages. Compared with CycleGAN, our method achieves the second goal better with a small sacrifice of the first goal. CycleGAN performs the translation to solely achieve the first goal. While the translated CTs by CycleGAN are more visually similar to the real CTs, they are not necessarily good for predicting the physiological age. And the prediction task is ultimately what we care about.

## 5 CONCLUSION

In this paper, we propose discriminative unpaired image-to-image translation, which translates images from source modalities into a target modality and use these translated images as augmented data for training the predictive model in the target modality. In our method, the training of the image-to-image translation module and the learning of the predictive module are conducted jointly so that the supervised information in the predictive task can guide the translation model. We apply our method for physiological age prediction. Experiments on three datasets demonstrate the effectiveness of our method.

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

APPENDIX

## A DETAILS OF EXPERIMENTS

### A.1 EXPERIMENT 1

#### A.1.1 DESCRIPTION

In this experiment, we train the model to generate fake CT images.

#### A.1.2 HYPERPARAMETER SETTINGS

| Name | Value | Description |
|------|-------|-------------|
| Optimizer | Adam | Algorithm to update model |
| Epoch | 200 | Number of total epochs to run |
| Learning rate (translator) | 2e-4 | Initial learning rate of translator |
| Learning rate (predictor) | 1e-3 | Initial learning rate of predictor |
| $\lambda_{cyc}$ | 10 | Tradeoff parameter for cycle consistency loss |
| $\lambda$ | 1e-3 | Tradeoff parameter between translator and predictor |
| Batch size | 16 | Batch size of all GPUs |
| Weight decay | 0 | Factor to regularize the model |

#### A.1.3 RUNTIME & COMPUTING INFRASTRUCTURE

It took about 60 hours to train the model to translate MRI images to CT images on GeForce GTX 1080 Ti ×4. It took about 110 hours to train the model to translate PET images to CT images on GeForce GTX 1080 Ti ×4. It took about 160 hours to train the model to translate MRI and PET images to CT images on TITAN Xp ×4.

### A.2 EXPERIMENT 2

#### A.2.1 DESCRIPTION

In this experiment, we train the model on pure CT images.

#### A.2.2 HYPERPARAMETER SETTINGS

| Name | Value | Description |
|------|-------|-------------|
| Model | Resnet50 | Model for training |
| Optimizer | Adam | Algorithm to update model |
| Epoch | 100 | Number of total epochs to run |
| Learning rate | 1e-4 | Initial learning rate |
| Batch size | 64 | Batch size of all GPUs |
| Weight decay | 0 | Factor to regularize the model |

#### A.2.3 RUNTIME & COMPUTING INFRASTRUCTURE

It took about 1 hours to train the model on GeForce GTX 1080 Ti ×1 .

### A.3 EXPERIMENT 3

#### A.3.1 DESCRIPTION

In this experiment, we train the model on MIXTURE of MRI and CT images.

| Name | Value | Description |
|---|---|---|
| Model | Resnet50 | Model for training |
| Optimizer | Adam | Algorithm to update model |
| Epoch | 100 | Number of total epochs to run |
| Learning rate | 1e-4 | Initial learning rate |
| Batch size | 64 | Batch size of all GPUs |
| Weight decay | 0 | Factor to regularize the model |

### A.3.2 HYPERPARAMETER SETTINGS

### A.3.3 RUNTIME & COMPUTING INFRASTRUCTURE

It took about 2 hours to train the model on GeForce GTX 1080 Ti $\times 1$ .

## A.4 EXPERIMENT 4

### A.4.1 DESCRIPTION

In this experiment, we train the model on MIXTURE of PET and CT images.

### A.4.2 HYPERPARAMETER SETTINGS

| Name | Value | Description |
|---|---|---|
| Model | Resnet18 | Model for training |
| Optimizer | Adam | Algorithm to update model |
| Epoch | 100 | Number of total epochs to run |
| Learning rate | 1e-4 | Initial learning rate |
| Batch size | 64 | Batch size of all GPUs |
| Weight decay | 0 | Factor to regularize the model |

### A.4.3 RUNTIME & COMPUTING INFRASTRUCTURE

It took about 3 hours to train the model on GeForce GTX 1080 Ti $\times 1$ .

## A.5 EXPERIMENT 5

### A.5.1 DESCRIPTION

In this experiment, we train the model on MIXTURE of MRI, PET and CT images. To stabilize the training, the model is pretrained on pure CT images.

### A.5.2 HYPERPARAMETER SETTINGS

| Name | Value | Description |
|---|---|---|
| Model | Resnet18 | Model for training |
| Optimizer | Adam | Algorithm to update model |
| Epoch | 100 | Number of total epochs to run |
| Learning rate | 1e-4 | Initial learning rate |
| Batch size | 64 | Batch size of all GPUs |
| Weight decay | 0 | Factor to regularize the model |

### A.5.3 RUNTIME & COMPUTING INFRASTRUCTURE

It took about 3 hours to train the model on GeForce GTX 1080 Ti $\times 1$ .

### A.6  EXPERIMENT 6

#### A.6.1  DESCRIPTION

In this experiment, the baseline method transfer learning(TL) is applied to MRI and CT images. The hyperparameter settings for pretraining and finetuning are the same.

#### A.6.2  HYPERPARAMETER SETTINGS

| Name | Value | Description |
|------|-------|-------------|
| Model | Resnet50 | Model for training |
| Optimizer | Adam | Algorithm to update model |
| Epoch | 100 | Number of total epochs to run |
| Learning rate | 1e-4 | Initial learning rate |
| Batch size | 64 | Batch size of all GPUs |
| Weight decay | 0 | Factor to regularize the model |

#### A.6.3  RUNTIME & COMPUTING INFRASTRUCTURE

It took about 1 hours to train the model on GeForce GTX 1080 Ti $\times 1$ .

### A.7  EXPERIMENT 7

#### A.7.1  DESCRIPTION

In this experiment, the baseline method Domain-Adversarial Neural Network (DANN) (Ganin & Lempitsky, 2015) is applied to MRI and CT images.

#### A.7.2  HYPERPARAMETER SETTINGS

| Name | Value | Description |
|------|-------|-------------|
| Optimizer | Adam | Algorithm to update model |
| Epoch | 100 | Number of total epochs to run |
| Learning rate | 1e-5 | Initial learning rate |
| Batch size | 64 | Batch size of all GPUs |
| Weight decay | 1e-3 | Factor to regularize the model |

#### A.7.3  RUNTIME & COMPUTING INFRASTRUCTURE

It took about 1 hours to train the model on GeForce GTX 1080 Ti $\times 1$ .

### A.8  EXPERIMENT 8

#### A.8.1  DESCRIPTION

In this experiment, the baseline method multitask learning (MTL) is applied to MRI and CT images. To stabilize the training, the model is pretrained on pure CT images.

#### A.8.2  HYPERPARAMETER SETTINGS

#### A.8.3  RUNTIME & COMPUTING INFRASTRUCTURE

It took about 2 hours to train the model on GeForce GTX 1080 Ti $\times 1$ .

| Name | Value | Description |
|------|-------|-------------|
| Model | Resnet50 | Model for training |
| Optimizer | Adam | Algorithm to update model |
| Epoch | 100 | Number of total epochs to run |
| Learning rate | 1e-4 | Initial learning rate |
| Batch size | 32 | Batch size of all GPUs |
| Weight decay | 0 | Factor to regularize the model |
| $\lambda_{mri}$ | 0.05 | Factor for task1(training MRI images) |
| $\lambda_{ct}$ | 0.95 | Factor for task2(training CT images) |

### A.9 EXPERIMENT 9

#### A.9.1 DESCRIPTION

In this experiment, fake CT images translated from MRI by CycleGAN (Zhu et al., 2017a) are combined with real CT images to train the model.

#### A.9.2 HYPERPARAMETER SETTINGS

| Name | Value | Description |
|------|-------|-------------|
| Model | Resnet50 | Model for training |
| Optimizer | Adam | Algorithm to update model |
| Epoch | 100 | Number of total epochs to run |
| Learning rate | 1e-4 | Initial learning rate |
| Batch size | 64 | Batch size of all GPUs |
| Weight decay | 0 | Factor to regularize the model |

#### A.9.3 RUNTIME & COMPUTING INFRASTRUCTURE

It took about 2 hours to train the model on GeForce GTX 1080 Ti $\times 1$ .

### A.10 EXPERIMENT 10

#### A.10.1 DESCRIPTION

In this experiment, the baseline method transfer learning(TL) is applied to PET and CT images. The hyperparameter settings for pretraining and finetuning are the same.

#### A.10.2 HYPERPARAMETER SETTINGS

| Name | Value | Description |
|------|-------|-------------|
| Model | Resnet50 | Model for training |
| Optimizer | Adam | Algorithm to update model |
| Epoch | 100 | Number of total epochs to run |
| Learning rate | 1e-4 | Initial learning rate |
| Batch size | 32 | Batch size of all GPUs |
| Weight decay | 0 | Factor to regularize the model |

#### A.10.3 RUNTIME & COMPUTING INFRASTRUCTURE

It took about 3 hours to train the model on GeForce GTX 1080 Ti $\times 1$ .

### A.11 EXPERIMENT 11

#### A.11.1 DESCRIPTION

In this experiment, the baseline method domain adaptation(DA) is applied to PET and CT images.

### A.11.2 HYPERPARAMETER SETTINGS

| Name | Value | Description |
|---|---|---|
| Optimizer | Adam | Algorithm to update model |
| Epoch | 100 | Number of total epochs to run |
| Learning rate | 1e-4 | Initial learning rate |
| Batch size | 32 | Batch size of all GPUs |
| Weight decay | 0 | Factor to regularize the model |

### A.11.3 RUNTIME & COMPUTING INFRASTRUCTURE

It took about 3 hours to train the model on GeForce GTX 1080 Ti $\times 1$ .

## A.12 EXPERIMENT 12

### A.12.1 DESCRIPTION

In this experiment, the baseline method multitask learning (MTL) is applied to PET and CT images. To stabilize the training, the model is pretrained on pure CT images.

### A.12.2 HYPERPARAMETER SETTINGS

| Name | Value | Description |
|---|---|---|
| Model | Resnet50 | Model for training |
| Optimizer | Adam | Algorithm to update model |
| Epoch | 100 | Number of total epochs to run |
| Learning rate | 1e-4 | Initial learning rate |
| Batch size | 32 | Batch size of all GPUs |
| Weight decay | 0 | Factor to regularize the model |
| $\lambda_{mri}$ | 0.1 | Factor for task1(training MRI images) |
| $\lambda_{ct}$ | 0.9 | Factor for task2(training CT images) |

### A.12.3 RUNTIME & COMPUTING INFRASTRUCTURE

It took about 4 hours to train the model on GeForce GTX 1080 Ti $\times 1$ .

## A.13 EXPERIMENT 13

### A.13.1 DESCRIPTION

In this experiment, fake CT images translated from PET by CycleGAN (Zhu et al., 2017a) are combined with real CT images to train the model.

### A.13.2 HYPERPARAMETER SETTINGS

| Name | Value | Description |
|---|---|---|
| Model | Resnet18 | Model for training |
| Optimizer | Adam | Algorithm to update model |
| Epoch | 100 | Number of total epochs to run |
| Learning rate | 1e-4 | Initial learning rate |
| Batch size | 64 | Batch size of all GPUs |
| Weight decay | 0 | Factor to regularize the model |

### A.13.3 RUNTIME & COMPUTING INFRASTRUCTURE

It took about 3 hours to train the model on GeForce GTX 1080 Ti $\times 1$ .

### A.14 EXPERIMENT 14

#### A.14.1 DESCRIPTION

In this experiment, basic data augmentation methods are applied to train the model.

#### A.14.2 HYPERPARAMETER SETTINGS

| Name | Value | Description |
|---|---|---|
| Model | Resnet50 | Model for training |
| Optimizer | Adam | Algorithm to update model |
| Epoch | 100 | Number of total epochs to run |
| Learning rate | 1e-3 | Initial learning rate |
| Batch size | 64 | Batch size of all GPUs |
| Weight decay | 0 | Factor to regularize the model |

#### A.14.3 RUNTIME & COMPUTING INFRASTRUCTURE

It took about 1 hours to train the model on GeForce GTX 1080 Ti $\times 1$ .

### A.15 EXPERIMENT 15

#### A.15.1 DESCRIPTION

In this experiment, AutoAugment method is applied to train the model.

#### A.15.2 HYPERPARAMETER SETTINGS

| Name | Value | Description |
|---|---|---|
| Model | Resnet18 | Model for training |
| Optimizer | Adam | Algorithm to update model |
| Epoch | 100 | Number of total epochs to run |
| Learning rate | 1e-4 | Initial learning rate |
| Batch size | 32 | Batch size of all GPUs |
| Weight decay | 0 | Factor to regularize the model |

#### A.15.3 RUNTIME & COMPUTING INFRASTRUCTURE

It took about 1 hours to train the model on GeForce GTX 1080 Ti $\times 1$ .

### A.16 EXPERIMENT 16

#### A.16.1 DESCRIPTION

In this experiment, random erasing method is applied to train the model.

#### A.16.2 HYPERPARAMETER SETTINGS

| Name | Value | Description |
|---|---|---|
| Model | Resnet18 | Model for training |
| Optimizer | Adam | Algorithm to update model |
| Epoch | 100 | Number of total epochs to run |
| Learning rate | 1e-4 | Initial learning rate |
| Batch size | 64 | Batch size of all GPUs |
| Weight decay | 0 | Factor to regularize the model |

### A.16.3 RUNTIME & COMPUTING INFRASTRUCTURE

It took about 1 hours to train the model on GeForce GTX 1080 Ti $\times 1$ .

## B ADDITIONAL EXPERIMENTAL RESULTS

Figure 4 shows more medical images from different modalities (i.e., CT, MRI and PET). Some fake CT images are generated from other modalities using our method (DUIIT) and CycleGAN (Zhu et al., 2017a).

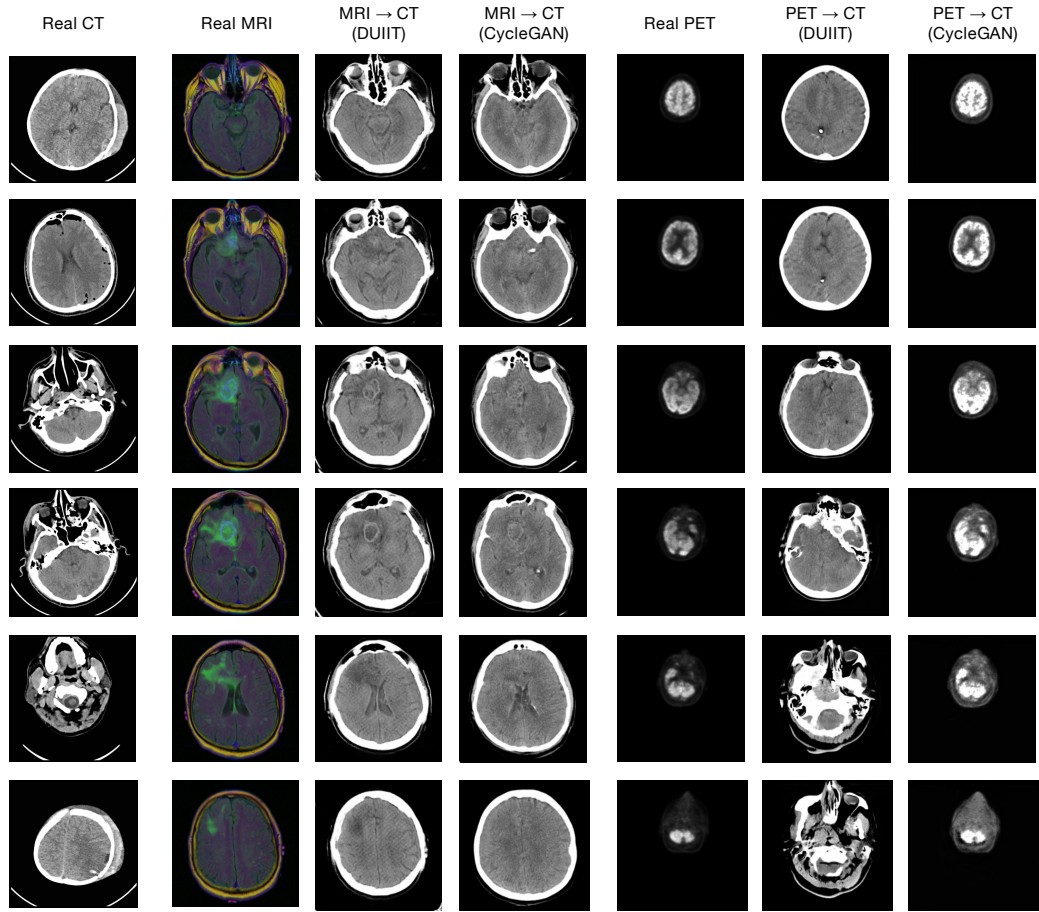

Figure 4: More real and fake images from different modalities(CT, MRI and PET). The fake images are generated using our method (DUIIT) and CycleGAN.

## C SIGNIFICANCE TEST BETWEEN METHODS

The significance test based on student t test is done between results gain from our methods and other methods. The significance threshold is set to 0.1. Table 6 shows the test results between MIX-MRI-CT and other methods. Table 7 shows the test results between MIX-PET-CT and other methods.

Table 6: Significance test between MIX-MRI-CT and other methods

|  | PURE-CT | TL | MTL | DA | CycleGAN | SimpleAugment | AutoAugment | RandomErasing |
|---|---|---|---|---|---|---|---|---|
| P-value | 0.020 | 0.017 | 0.000 | 0.010 | 0.013 | 0.013 | 0.000 | 0.089 |
| Different distribution or not | Yes | Yes | Yes | Yes | Yes | Yes | Yes | Yes |

Table 7: Significance test between MIX-PET-CT and other methods

|  | PURE-CT | TL | MTL | DA | CycleGAN | SimpleAugment | AutoAugment | RandomErasing |
|---|---|---|---|---|---|---|---|---|
| P-value | 0.026 | 0.001 | 0.015 | 0.000 | 0.003 | 0.025 | 0.000 | 0.043 |
| Different distribution or not | Yes | Yes | Yes | Yes | Yes | Yes | Yes | Yes |

