# OpenReview forum: "Discriminative Cross-Modal Data Augmentation for Medical Imaging Applications"
_ICLR.cc/2021/Conference — Reject_

### Official Review · AnonReviewer1 · 2020-10-25
**Nice results, somewhat limited novelty**

**Rating:** 5
**Confidence:** 4

**Review:**

The authors propose an algorithm to enlarge the training set for image classification problems in certain medical applications where training data of the target modality is scarce. They do so by training an unpaired image-to-image translator network and an image classifier end-to-end in order to utilize labeled images acquired through various imaging modalities. Moreover, they demonstrate the effectiveness of the proposed algorithm via extensive numerical experiments on a prediction problem and compare their method to several different approaches ranging from transfer learning to data augmentation.

Positives:
+ The problem is well-motivated. Obtaining large medical image datasets is difficult, therefore it is very important to find ways to use the available data as efficiently as possible.
+ In most parts, the paper is well-written, clear and easy to follow.
+ Authors compared their method to a wide range of other possible techniques.
+ The experimental results are very promising for the given prediction problem.

Negatives:
- The novelty of the paper seems to be somewhat limited. Using unpaired image-to-image translation networks for cross-modal medical image synthesis has been studied in [1, 2] (as the authors pointed out in the paper). The only novelty compared to [2] seems to be that authors apply the idea to an image classification problem instead of segmentation. Authors claim that part of the novelty is that in their method translation is guided by the predictive task, however in [2] the translator is also jointly trained with the segmentation network.
- The consistency of anatomical  structures is not enforced in any way in the translated images (see shape consistency in [2]). For instance, CT images generated from PET are very dissimilar to original PET images (Fig. 2). This might lead to loss of features crucial for diagnosing certain diseases, or even to false diagnosis via 'hallucinated' features. Medical diagnosis is often made based on very specific, fine features on the image. The experiments performed in the paper pertain to physiological age prediciton where specific fine-grained features might not be necessary for correct prediction. Therefore further experiments might be needed to judge the wider applicability of the method.
- There is some confusion around the experimental results. First, in Table 3 some results (CycleGAN for MRI-CT and DA for PET-CT) of competing methods are worse than simply training on the CT images (PURE-CT in Table 2). Second, based on the Appendix, in some experiments the backbone model is ResNet18 and in other experiments it is ResNet50. The data augmentation experiments from Table 4 were performed using ResNet-18, whereas for the proposed method ResNet50 has been used. It is not clear how the results are comparable given the difference in network capacity.

Even though the paper shows promise, given the limited novelty and somewhat questionable applicability of the presented results as detailed above, in this form I would recommend rejection.

Could the authors clarify what the main contribution of the paper is compared to other techniques exploiting cross-modal image synthesis for medical applications. Furthermore, could the authors comment on my concerns about geometric distortions originating in image translation and how it may effect medical diagnosis? Lastly, please address the issues raised about the experimental results above.

Additionally, I have some minor comments and suggestions:
- The algorithm could be better differentiated from other results in the related work section.
- It is not clear how the discussion on text-to-image generation methods in Section 2.2 is relevant to the paper.
- Using the same colormap for all figures for different modalities would be helpful for better comparison.
- It is difficult to match the experimental details in the appendix to the experiments in the paper.
- There is a typo in the title of Section 3.3

[1] Chartsias et al, Adversarial image synthesis for unpaired multi-modal cardiac data, International workshop on simulation and synthesis in medical imaging, 2017

[2] Zhang et al, Translating and segmenting multimodal medical volumes with cycle-and shape-consistency generative adversarial network, CVPR, 2018

Post-rebuttal: The authors did not provide feedback and therefore I keep my score.

---

### Official Review · AnonReviewer3 · 2020-10-28
**Good direction but lacks technical novelty**

**Rating:** 4
**Confidence:** 4

**Review:**

Summary:
This paper studies the problem of learning a predictor from a specified modality using a dataset where each example has images from only one modality. The proposed approach is to set up a CycleGAN to translate between the modalities and a predictor from the required modality to the prediction target. The authors propose to learn the CycleGAN and the predictor parameters jointly. Experimental results show that this performs better than either not doing the translation at all (i.e. only learning the predictor from examples with the required modality) or learning the translation followed by learning the prediction.


Strengths:
+ Interesting problem setting. I find the medical imaging use case convincing.
+ The proposed idea is simple and clean
+ Empirical evaluation uses a comprehensive set of baselines


Concerns / weaknesses:
- The technical novelty in this paper is unclear. It has been generally known that an end-to-end approach typically works better than learning the two parts (CycleGAN-translator and predictor) separately.
- Authors state that they randomly split images into training, validation and test sets. However, since there are ~30 images per patient, this could result in images from the same patient ending up in training as well as test sets. I imagine nearby slices of the 3D volume represented by a single CT-scan to be very similar to each other, and so this raises a serious concern about train/test overlap.
- Related to above (and even after using per-patient splits), the std-dev of the prediction errors needs to be estimated using independent samples (i.e. from different patients), since samples from the same patients would not be independent. It is unclear how these correlations were handled.

---

### Official Review · AnonReviewer2 · 2020-10-29
**Official Blind Review #2**

**Rating:** 5
**Confidence:** 4

**Review:**

This paper combines the Cycle-GAN with the predictor for downstream task to augment the training data in the target modality. On two applications, the experimental results illustrate the effectiveness of their method.

Strengths:
(1) The paper is very well-written and easy to be understood.
(2) Compared with the baselines, the proposed method significantly improves the performance of downstream regression task.

Weakness:
(1) The comparison between the baselines and proposed method is not fair. For example, Data augmentations methods only use the target dataset, but the proposed method use both the source and target datasets.
(2) The combination between the Cycle-GAN and downstream predictor is not used at first. [1] has introduced the segmentor to constrain the generated images. The difference is that the proposed method is applied to regression task and [1] solved the segmentation task.

[1] Zhang, Zizhao, Lin Yang, and Yefeng Zheng. "Translating and segmenting multimodal medical volumes with cycle-and shape-consistency generative adversarial network." Proceedings of the IEEE conference on computer vision and pattern recognition. 2018..

---

### Official Review · AnonReviewer4 · 2020-10-31
**interesting approach**

**Rating:** 6
**Confidence:** 5

**Review:**

Summary:
This paper discusses an approach to augment a medical imaging dataset using images from another modality. The images in the other (i.e. source) modality should have been originally collected & labeled to perform the same discrimination or regression task as the target modality. A network consisting of a prediction network for the mixture of augmented and target images & another network based on CycleGAN for the image translation network are jointly optimized using end to end training.

Reasons for score:
The objective of the paper (i.e. addressing small training set size in medical imaging) is quite important, and the approach is interesting. However, the evaluation, discussion, and generalizability of the approach to other tasks require additional clarification, simulations, and extra information.

Detailed review:
1) Is physiological age in the clinic estimated from a single slice in a CT volume, or an entire volume? If latter, the entire evaluation strategy should change to include the set of images from the volume rather than single images. Otherwise, the task selected will not have any bearing on the actual clinical task of estimation of physiological age. This would also impact the approach which uses a 2D image to image translation, as it will likely mean that a 3d to 3d image translation should be used.
2) The authors have not provided sufficient information about the distribution of attributes of the different datasets. This information is critical in assessing the generalizability of results, as well as whether the experiments were set up in a meaningful way.
a) What is the distribution of actual age in the different datasets from the different modalities?
b) What are the image acquisition / reconstruction attributes of source/target datasets? Images from the same patient will look very different and contain varying degrees of anatomical detail depending on slice thickness, dose in CT & imaging sequence in MRI, reconstruction algorithm, etc. When doing image to image translation, does e.g. slice thickness across the modalities need to match each other?
3) In medical imaging we often have severe class imbalance, where disease positive samples are much more rare compared with disease negative samples. Given that datasets from different modalities will likely have different ratios of positive to negative samples, how will this affect the overall training strategy? E.g. would the authors only augment the positives in this way , or both positives & negatives? Otherwise, if images from source modality have different class balance than the one in target?
4) The proposed approach has been described for a medical imaging task that involves macro anatomical features only. It is therefore not clear whether this approach would generalize to a task that pertains to more micro features (e.g. tumor classification/detection, or tumor segmentation). Based on Fig2, it appears that the MRI images converted to CT contain obvious anatomical inaccuracies (e.g. Fig 4 shows consistently thick skulls in the generated CT images, which would affect a brain volume estimation task). Have the authors used this approach for other tasks that require more micro level anatomical precision? If not, this should be explicitly stated as a shortcoming of the current approach.
5) The selection of hyper-parameters, experimental setups, and split of data into train/test using different approaches requires more explanation:
a) It is not clear how the lambda values were selected. Also, for eq 3 the authors state that a lambda of 0.001 was used, which seems to severely favor the GAN loss rather than the prediction loss. Given that the prediction task is the more important among the two, no justification has been provided on why the weight of the corresponding loss value would be so small compared to the GAN loss.
b) What was the stopping criteria for the different scenarios in Table 2? In the Appendix, the authors have the number of training epochs for each approach (which is different for different experiments), but it is not clear what determined the end of training. This is important, since all approaches should be trained using the same rule to ensure a fair comparison.
c) On p5, the authors state that they randomly select images from the CT cases to split the data into train/val/test buckets; Does this mean that images from same patient are mixed into both train/test? This would not be appropriate, since the different slices of CT data from the same patient share anatomical similarity, which means that the train/test data are likely cross-contaminated.
d) In Table 2, what error is being shown? Is it l1 norm, l2 norm, or absolute error of predicted age relative to annotated age?
e) What is the actual loss function in eq 2? The loss for L_adv has been described on p6 but not the loss in eq2. Also, predicting age is a regression task, and not a discrimination task. Why is eq2 described as showing a discrimination task?
6) In Table 3, why is the cyclegan result  significantly worse than other methods for MRI (it actually deteriorates the performance compared to baseline)? On the other hand for PET, cyclegan is better than other methods; The opposite trend is happening for domain adaptation (i.e. worse for PET and better for MRI). Is this related to the amount of training data? Otherwise, this would seem to indicate that perhaps cyclegan did not train properly due to implementation issues, etc.
7) In Figs2-4, visual comparison of the translated images to real CT images have been provided. However, since unpaired data was used, it is not clear how anatomically correct the translated images are. It would be best if for a small number of samples that have paired data (i.e. patient was imaged using both CT and MR, or CT and PET), the authors show a comparison of the translated images from MR or PET compared to the actual patient images in CT. Such a comparison would show beyond doubt that the translated images are anatomically correct or not.

---

### Decision · Program_Chairs · 2021-01-07
**Final Decision**

**Decision:**

Reject

**Comment:**

The paper proposes a method for data augmentation by cross-modal data generation. While the reviewers agree that the paper addresses a relevant and important problem in medical imaging, they also agree on that the paper has limited novelty over the state of the art. Also the setup of experimental validation to comparison methods is questioned.